# First Application of Nitrogen-Doped Carbon Nanosheets Derived from Lotus Leaves as the Electrode Catalyst for Li-CO$_2$/O$_2$ Battery

**Lu Zou [1],\*, Weilin Kong [1], Linfeng Peng [1] and Fang Wang [2,3],\***

[1] Key Laboratory of Green Chemical Engineering Process of Ministry of Education, School of Chemical Engineering and Pharmacy, Wuhan Institute of Technology, Wuhan 430073, China

[2] School of Materials Science and Engineering, Changchun University of Science and Technology, Changchun 130022, China

[3] Zhongshan Institute of Changchun University of Science and Technology, Changchun 130012, China

\* Correspondence: zoulu237@126.com (L.Z.); f-wang-ssi@foxmail.com (F.W.)

**Abstract:** The development of Li-CO$_2$/O$_2$ battery with high energy density and long-term stability is urgently needed to fulfill the carbon neutralization and pollution-free environment targets. The biomass-derived heteroatom-doped carbon catalyst with the combination of high-efficiency catalytic activity and sustainable supply is a promising cathode catalyst in Li-CO$_2$/O$_2$ battery. Specifically, the unique morphology and mesopore structure can promote the transfer of CO$_2$, O$_2$, and Li$^+$. Abundant channel pores can provide discharge products accommodation to the largest extent. Nitrogen dopant, the commonly recognized active sites in carbon, can improve the electron conductivity and accelerate the sluggish kinetic reaction. Therefore, utilizing the louts leaves as the precursor, we successfully prepare the cellular-like nitrogen-doped activated carbon nanosheets (N-CNs) through the appropriate pyrolysis carbonization method. The as-synthesized carbon nanosheets display a three-dimensional interconnecting pore structure and abundant N-dopant actives, which dramatically improve the electrochemical catalytic activity of N-CNs. The Li-CO$_2$/O$_2$ battery with the N-CNs cathode delivers a high discharge capacity of 9825 mAh g$^{-1}$, low overpotential of 1.21 V, and stable cycling performance of 95 cycles. Thus, we carry out a facile method for N-doped carbon nanosheets preparation derived from the cheap natural biomass, which can be the effective cathode catalyst for environmental-friendly Li-CO$_2$/O$_2$ battery.

**Keywords:** biomass-derived carbon; carbon nanosheets; nitrogen defects; cathode catalyst; Li-CO$_2$/O$_2$ battery

## 1. Introduction

The rapidly increasing carbon dioxide (CO$_2$) emissions caused by the overuse of fossil fuel make us urgently search for the renewable energy storage and conversion devices to realize the environmentally sustainable development [1]. Li-CO$_2$/O$_2$ battery, a promising and innovative energy storage strategy, largely attracts researchers' attention for its combination of CO$_2$ capture and electrical energy storage and generation [2,3]. Similar with the traditional Li-O$_2$ battery, a typical Li-CO$_2$/O$_2$ battery consists of lithium anodes, electrolyte, and porous cathodes conducive to CO$_2$/O$_2$ adsorption and diffusion, which can deliver higher theoretical energy density than that of traditional Li-O$_2$ battery based on the reaction mechanisms (4Li$^+$ + 2CO$_2$ + O$_2$ + 4e → 2Li$_2$CO$_3$) [4]. However, the development of Li-CO$_2$/O$_2$ battery is still only in its infancy, hindered by inevitable issues, such as low practical discharge capacity, poor rate capability, high overpotential, and poor cyclability, which are caused by the sluggish Li$_2$CO$_3$ aggregation and decomposition kinetics [5–8]. In this regard, effective cathode with desirable catalytic activity promoting Li$_2$CO$_3$ deposition/decomposition and cathode structure improving gas diffusion,

Li$_2$CO$_3$ aggregation, and electronic conductivity, should be effectively designed for high-performance Li-CO$_2$/O$_2$ battery.

Up to now, extensive cathode catalyst has been developed to facilitate the reversible formation and decomposition of Li$_2$CO$_3$, for the purpose of fulfilling the Li-CO$_2$/O$_2$ battery with high efficiency and cycle stability [9]. Similar to the Li-O$_2$ battery, the noble metal and transition metal oxides have been extensively utilized as efficient catalysts in Li-CO$_2$/O$_2$ battery [10]. However, critical disadvantages, such as high-cost, scarcity, and limited catalytic efficiency hinder the commercialization of Li-CO$_2$/O$_2$ battery. More importantly, the large numbers of essential heavy metals and ions will cause detrimental effects on the environmental safety. Therefore, considerable research on the carbon-based electrocatalysts has been reported in Li-CO$_2$/O$_2$ battery. Carbon nanotubes (CNTs) and graphene, typical carbon catalysts, have been applied as cathode materials in Li-CO$_2$/O$_2$ batteries, which are demonstrated to promote the formation and decomposition of Li$_2$CO$_3$ [10–12]. However, the catalytic efficiency and stability of the carbon catalyst need further improvement. Appropriate heteroatom-doped carbon or synergistic effect are considered to improve the catalytic activity. Li et al. [13] prepared a highly efficient Pt/FeNC catalyst for high-performance rechargeable Li-CO$_2$/O$_2$ batteries. Utilizing the synergetic effect of Pt NPs (2.4 nm) and 2D Fe-N-C matrix derived from MOFs, the Li-CO$_2$/O$_2$ battery displayed a low overpotential of 0.54 V and stable cycling performance of 142 cycles. Zhang et al. [14] realized a uniform dispersion of Pt NPs (2.5 nm) on carbon nanotubes (Pt/CNT) in Li-CO$_2$/O$_2$ battery. Benefiting from the Pt/CNT structure, a valuable Li$_2$CO$_3$ deposition inside Pt/CNT layer was accelerated. Consequently, the related Li-CO$_2$/O$_2$ battery (20% O$_2$) exhibited a low overpotential of 0.51 V and cycling stability of 90 cycles. Sun et al. [15] synthesized a core-shell Ru/NiO@Ni/CNT catalyst with Ru NPs (~2.5 nm) anchoring on core-shell-like NiO@Ni, utilizing CNT as catalyst support. For the strong interfacial interactions between Ru NPs and NiO, the corresponding Li-CO$_2$/O$_2$ batteries could enable low voltage hysteresis (1.01 V) and long cycle life (105 cycles). Furthermore, the Ru/N-doped carbon nanotube (Ru/NC) catalyst was also employed in Li-CO$_2$/O$_2$ battery. Due to the superior catalytic activity of Ru NPs and N-doped carbon, the decomposition kinetics of Li$_2$CO$_3$ were successfully accelerated and the fabricated Li-CO$_2$/O$_2$ battery performed acceptable electrochemical performance with low overpotential of 1.06 V and cycling stability of 90 cycles [16]. Apparently, the heteroatom-doped carbon especially the N-doped carbon catalyst can be considered as the potential cathode for high efficiency Li-CO$_2$/O$_2$ battery.

Meanwhile, the biomass-derived heteroatom-doped carbon material has been considered as the environmentally friendly and inexpensive cathode catalyst for Li-CO$_2$/O$_2$ battery [17–24]. Specifically, the unique morphology property, abundant microchannel structure, sufficient conductivity, and considerable specific surface area can largely facilitate the CO$_2$, O$_2$, Li$^+$, and electron transfer, electrolyte immersion, and discharge product accumulations. Therefore, extensive efforts have been devoted to utilizing these waste resources as the efficient energy carriers. Yang et al. [17] prepared a novel three-dimensional binder-free N-doped carbon nanonet adopting silkworm cocoon as the precursor. The assembled Li-O$_2$ battery delivered a high specific capacity of 1480 mAh g$^{-1}$ and acceptable cycling performance of 60 cycles at 0.25 mA cm$^{-2}$. Huang et al. [24] successfully synthesized the N-doped activated carbons (N-PIACs) derived from poplar inflorescence, which presented a three-dimensional interconnecting pore structure and owned defects and functional groups by N-doping. Correspondingly, the enhanced electrochemical activity of Li-O$_2$ battery was achieved with a high specific capacity of 12,060 mAh g$^{-1}$ and stable cycling stability of 86 cycles. Zhang et al. [19] adopted the citrus maxima peel as a precursor for the cellular-like carbon catalyst (CMPACs). The Li-O$_2$ battery with CMPAC cathode possessed high specific capacity of 7800 mAh g$^{-1}$, excellent cycling performance of 466 cycles, as well as good coulombic efficiency of 92.5%. Zhu et al. [22] developed a wood-derived, free-standing porous carbon electrode in Li-O$_2$ battery. Benefiting from the spontaneously formed hierarchical porous structure and N dopant, the catalytic activity of the carbon cathode was improved with lower overpotential and higher capacity.

Lotus leaves, the most available green and sustainable biomass in China, have provided a high ecological and economic value in the biotechnology, medicine, and function fiber, as well as energy storage and conversion areas [25–29]. Deng et al. [28] prepared hydrophilic porous carbon nanosheet using lotus leaf as the carbon source. When adopted as the electrode in supercapacitor, the lotus leaf derived carbon exhibited specific capacitances of 225 F $g^{-1}$ and 289 F $g^{-1}$ at 0.5 A $g^{-1}$ in 1 M NaCl and 6 M KOH electrolytes. Wu et al. [30] carefully investigated the $CO_2$ adsorption property and supercapacitor performance of N-doped porous carbons from lotus leaf, which possessed good $CO_2$ adsorption capacity of 3.50 mmol/g (25 °C) and 5.18 mmol/g (0 °C) under atmospheric pressure, and a high capacitance (266 F $g^{-1}$) was also achieved in supercapacitor. Ma et al. [31] successfully explored the lotus leaf as the low-cost carbon source in supercapacitor. In 6 M KOH electrolyte, the carbon catalyst exhibited a high specific capacitance of 379 F $g^{-1}$ and good rate performance at 1 A $g^{-1}$. Li et al. [32] prepared the biocarbon coated $Li_3V_2(PO_4)_3$ cathode material using lotus leaf as carbon source by sol gel method, which successfully improved the lithium-ion diffusion coefficient of $Li_3V_2(PO_4)_3$ through carbon layer. The corresponding Li-ion battery delivered a high initial discharge capacity of 130.4 mAh $g^{-1}$ and coulombic efficiency of 99% under 0.1 C. Zhou et al. [33] employed the lotus leaf with intrinsically hierarchical structure as the single precursor for oxygen reduction reaction (ORR) catalyst, which achieved an onset potential of −0.015 V vs. Ag/AgCl (commercial Pt/C catalyst: −0.010 V vs. Ag/AgCl). Inspired by the above research, the heteroatom-doped carbon synthesized by the lotus leaf can be considered as the high-efficiency catalyst in $Li-CO_2/O_2$ battery.

Herein, the lotus leaf is chosen as the precursor to prepare biomass-derived N-doped activated carbon nanosheets. With the assistant of pyrolysis and activation technology, the cellular-like carbon catalyst has been prepared with porous morphology, high surface area, and abundant active sites, which can largely improve the catalytic activity. The $Li-CO_2/O_2$ battery using N-doped carbon delivers a high discharge capacity of 9825 mAh $g^{-1}$, acceptable discharge-charge overpotential of 1.21 V, and long-term cycling stability of 95 cycles. The results strongly prove that the biomass-derived carbon catalyst can be the alternative candidate for high-performance cathode catalyst in $Li-CO_2/O_2$ battery. More importantly, multiple advantages of this approach, including, but not limited to, high efficiency and facile preparation, resource-unlimited raw material, and spontaneous unique morphology, will contribute to the development of economic and environmentally friendly $Li-CO_2/O_2$ battery.

## 2. Results and Discussion

Figure 1 shows the morphology and microstructure evolutions of lotus leaf under different pyrolysis conditions. When the chlorophyll precursor was calcinated at 400 °C for 1 h under the 50 mL $min^{-1}$ $N_2$ atmosphere, the as-synthesized carbon presents a three-dimensional framework covered by the dense film. When the temperature increases to 600 °C (Figure 1b), the three-dimensional pore structure with the average size of 1 μm can be directly observed, which may be derived from the organic compound decomposition. Furthermore, the walls existed between the inter-channel pores become much thinner at higher temperature of 800 °C, as shown in Figure 1c. It was found that the three-dimensional carbon framework is blocked by the mesoporous inter-channel pores with an average pore size of 20 nm, indicating the hierarchical pore structure. When the $N_2$ flow increases to 100 mL $min^{-1}$, numerous regular honeycomb holes consisting of carbon nanosheets were observed at 800 °C, which display a highly arranged mesopores structure in Figure 1d. As the $N_2$ flow continue to 150 mL $min^{-1}$, the honeycomb holes were found to be collapsed (Figure 1e) and eventually turned into the one consisting of disorganized carbon fibers and particles under the $N_2$ flow of 200 mL $min^{-1}$ (Figure 1f).

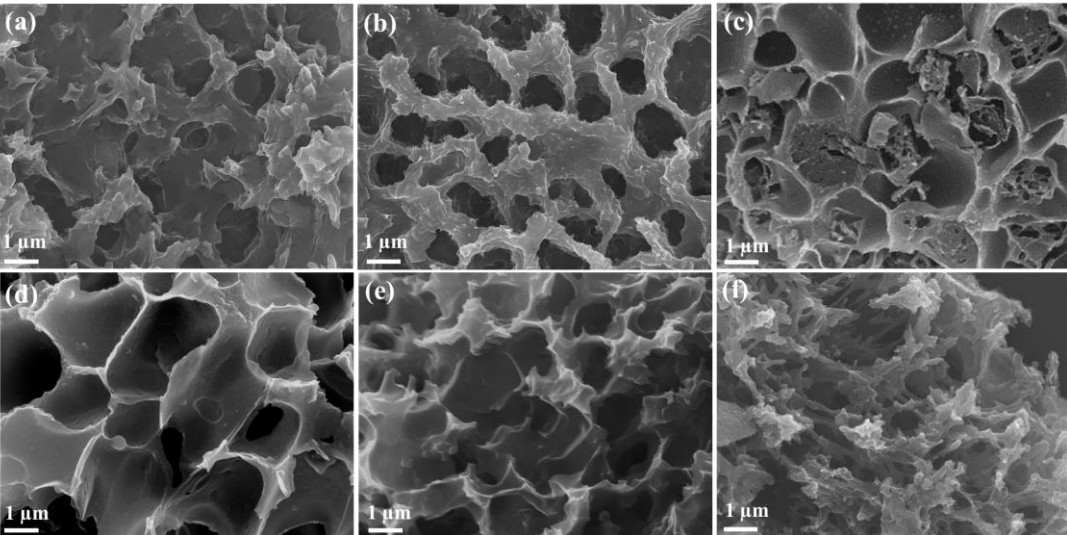

**Figure 1.** SEM images of the activated lotus leaf at different calcination conditions (**a**) 400 °C, (**b**) 600 °C, (**c**) 800 °C under 50 mL min$^{-1}$ of $N_2$; 100 mL min$^{-1}$ (**d**), 150 mL min$^{-1}$ (**e**), and 200 mL min$^{-1}$ (**f**) of $N_2$ at 800 °C.

Figure 2 shows the detailed morphology characteristic of carbon nanosheets under the optimized pyrolysis conditions (800 °C, 100 mL min$^{-1}$). Apparently, the highly interconnected three-dimensional architecture stabilized by the networking of cellular-like carbon forms a mechanically robust framework with open pore structures, as well as abundant meso- and macro-pores (Figure 2a), which are caused by the emission of pressured gas demonstrated in Figure 2b. Moreover, the meso- and macro-pores play a key role in enabling electrolyte penetration into the hierarchical structure. As seen in Figure 2c,d, the ultrathin wall with an average thickness of 8 nm can be clearly observed along the carbon nanosheets edges. The distinct interlayer distance ascribing to carbon can further prove the formation of carbon nanosheets. Consequently, the ultrathin walls (8 nm) and abundant mesopores are directly confirmed, which can shorten the transport paths for $CO_2$, $O_2$, and $Li^+$, enlarge the accommodation for insoluble discharge product, provide numerous accessible catalytic sites toward $Li_2CO_3$ deposition and decomposition reactions, and eventually ensure the effective and stable catalytic activity.

Figure 3 shows the XRD patterns and Raman spectra of carbon nanosheets and Super P for confirming the formation of carbon phases. As shown in Figure 3a, the broad diffraction peaks observed at ~26.6° and 44.6° can be attributed to the (002) plane and (100) plane of amorphous carbon, respectively. Additionally, Raman spectra of the corresponding catalysts are obtained to analyze the detailed structural disorder of the as-synthesized carbon nanosheets, as shown in Figure 3b. Two remarkable Raman shift peaks, characteristics of G band (1580 cm$^{-1}$) and D band (1300 cm$^{-1}$) for the graphitic carbon, are observed for both carbon nanosheets and Super P. Among these, the D-band corresponds to sp$^3$ hybridized carbon with the disordered state, whereas the G-band reveals the planar vibration of the sp$^2$ carbon. Importantly, the disorder degree of the carbon materials can be quantified by the intensity ratio of D band to G band ($I_D/I_G$ ratio). With the fitting areas of the two typical curves, the $I_D/I_G$ ratio of carbon nanosheets is remarkably higher than that of Super P. The results demonstrate that the carbon nanosheets are preferably decorated with the defect sites compared with that of Super P. It could be speculated that the conductivity and catalytic activity of carbon nanosheets can be improved.

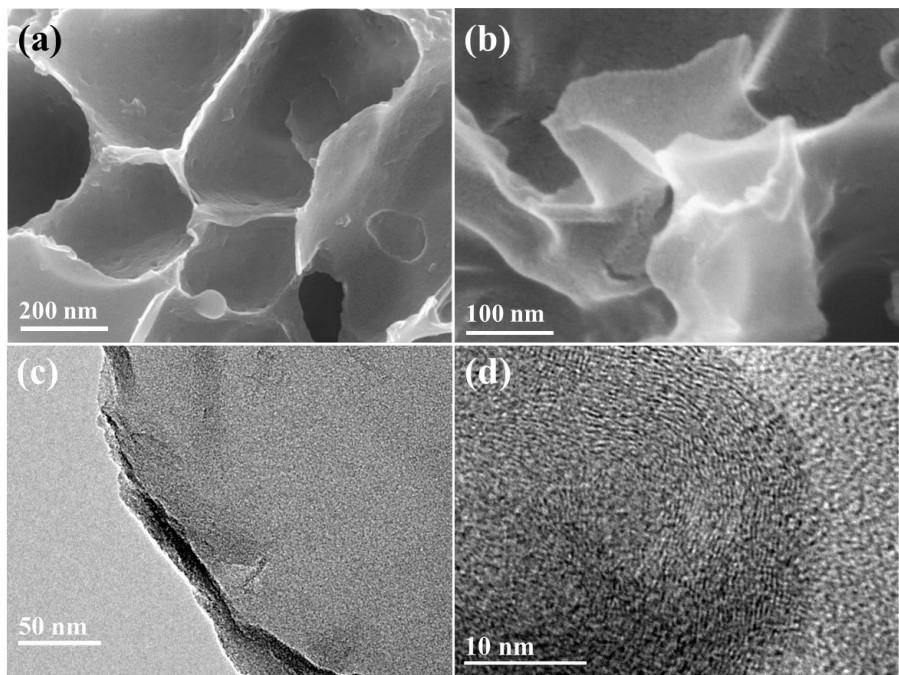

**Figure 2.** SEM (**a**–**c**) and TEM (**d**) images of the activated lotus leaf at 800 °C for 1 h with 100 mL min$^{-1}$ of $N_2$.

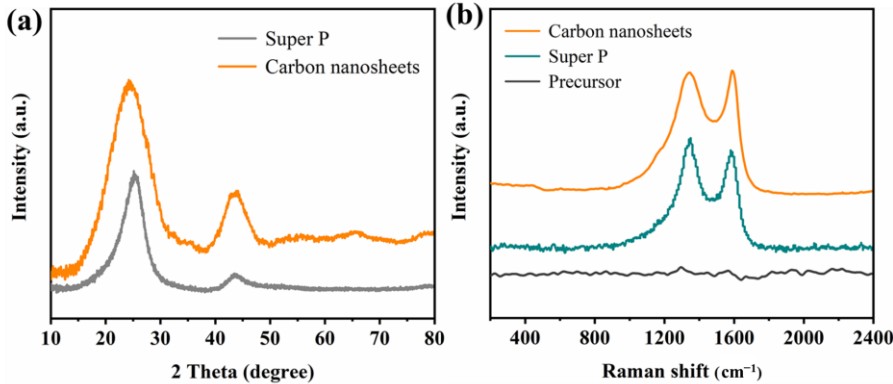

**Figure 3.** XRD (**a**) and Raman patterns (**b**) of carbon nanosheets and Super P.

Figure 4 provides quantitative evidence of the mesopores observation of synthesized carbon nanosheets through nitrogen adsorption–desorption isotherms. Commercial Super P is used as the benchmark, which has been extensively used as the conductive agent in Li-$CO_2$/$O_2$ battery. Clearly, carbon nanosheets exhibit type IV isotherms with obvious hysteresis loop, indicating a characteristic of mesopore structure, which is much different from the microporous Super P in the inset curves. A typical H1 hysteresis loop appearing at high P/$P_0$ range of 0.8–1.0 could be observed for Super P. Moreover, the carbon nanosheets with interconnected three-dimensional architecture presented a BET surface area of 116.3844 m$^2$ g$^{-1}$ and a narrow pore size distribution centered at 5 nm. In contrast, the commercial Super P has a BET surface area of 58 m$^2$ g$^{-1}$ and broader pore size distribution, an indicator of the micropore characteristics. For the synthesized carbon nanosheets, such mesoporosity is expected to provide abundant active sites, faster $CO_2$/$O_2$ transfer, and enlarged $Li_2CO_3$ accumulation.

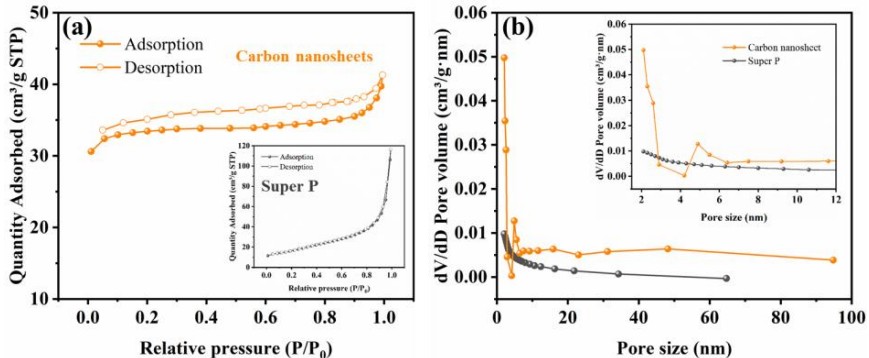

**Figure 4.** $N_2$ adsorption-desorption isotherms (**a**) and the pore size distribution (**b**) of the as-synthesized carbon nanosheets and Super P.

Figure 5 shows the elemental composition and oxidation states of carbon nanosheets by XPS survey. The XPS spectrum displays signals of C 1s peak (76.28 at%), O 1s (21.15 at%), and N 1s (2.57 at%), revealing the high O content and presence of structural N in the carbon nanosheets. Besides, the high-resolution C 1s spectrum can be divided into three peaks at 284.5 eV, 285.6 eV, and 288.3 eV, respectively, corresponding to the C–C, C=O, and –COO components. It is apparently observed that the high-resolution O 1s spectrum presents the coexistence of different function groups, including C=O (531.5 eV), C–O (532.0 eV), and –OH (533.4 eV) groups, respectively. Additionally, the fitted O 1s spectrum distinctly indicates the surface-bound oxygen is mainly C–O group. In addition, the N 1s spectrum can be divided into three peaks ascribing to pyridinic nitrogen, pyrrolic nitrogen, and graphitic nitrogen. Among these, pyridinic and pyrrolic nitrogen are confirmed to be the active sites for $CO_2$ adsorption, graphitic and pyridinic nitrogen are beneficial to the $O_2$ reduction process, and the pyrrolic nitrogen can enhance the adsorption between lithium ions and catalyst [22,23,34]. Thus, the coexistence of pyridinic, pyrrolic, and graphitic nitrogen in the carbon nanosheets could improve the electrochemical performance of the Li-$CO_2$/$O_2$ battery.

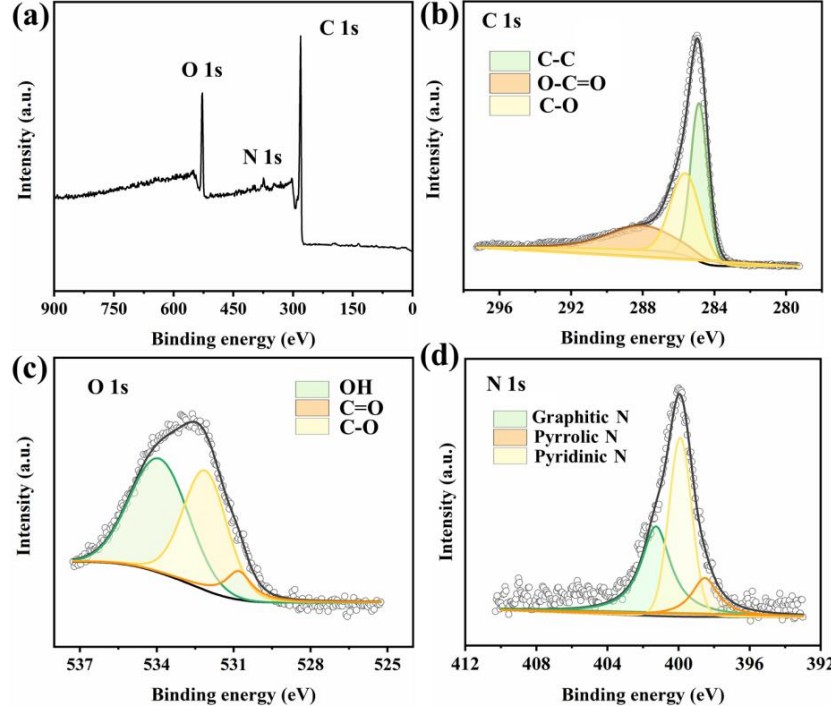

**Figure 5.** XPS total spectrum (**a**), C 1s spectrum (**b**), O1s spectrum (**c**) and N1s spectrum (**d**) of carbon nanosheets.

Figure 6 shows the electrocatalytic activity and electron conductivity of the carbon nanosheets and Super P catalysts, which are examined by CV and EIS measurements. As the benchmark catalyst, the Super P catalyst exhibits no catalytic activity in $CO_2/O_2$, as shown in Figure 6a. Surprisingly, the cathodic peak of carbon nanosheets is observed at 2.5 V, indicating the $O_2$ reduction process. In addition, the current density of the carbon nanosheets along anodic and cathodic direction are much higher than that of commercial Super P, revealing the considerable catalytic activity in Li-$CO_2/O_2$ battery. Besides, the electron conductivity of the battery with carbon nanosheets and Super P are further investigated in Figure 6b. Clearly, the Ohmic resistance (Ro) and charge transfer resistance (Rct) of carbon nanosheets are both lower than that of Super P cathode, revealing the remarkable enhanced electrical conductivity of carbon nanosheets.

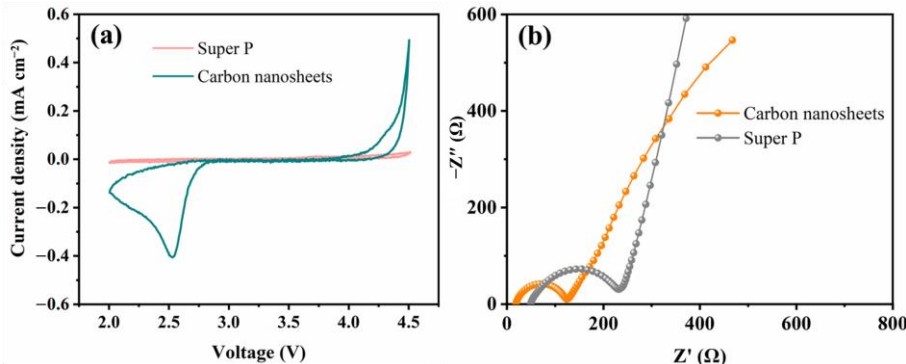

**Figure 6.** CV curves (**a**) and EIS spectrum (**b**) of carbon nanosheets and Super P catalysts.

Figure 7 shows the capacity performance and long-term cycling stability of the Li-$CO_2/O_2$ battery with carbon nanosheets and Super P catalysts. Firstly, the capacity performance of Li-$CO_2/O_2$ battery catalyzed by carbon nanosheets and Super P are shown in Figure 7a–c. The battery with carbon nanosheets cathode delivers the discharge capacity of 9825 mAh g$^{-1}$, 9078 mAh g$^{-1}$, 8295 mAh g$^{-1}$, 7264 mAh g$^{-1}$, 5841 mAh g$^{-1}$ at the current density of 200 mA g$^{-1}$, 500 mA g$^{-1}$, 800 mA g$^{-1}$, 1200 mA g$^{-1}$, and 1500 mA g$^{-1}$, revealing the outstanding capacity performance, excellent coulombic efficiency, and catalytic stability at high current density. Moreover, the comprehensive capacity performance of carbon nanosheets based Li-$CO_2/O_2$ battery is much better than that of the Super P cathode (the highest capacity of 8908 mAh g$^{-1}$ at 200 mA g$^{-1}$). The first discharge–charge cycle under different restricted conditions is collected in Figure 7d. The discharge platform of carbon nanosheets based Li-$CO_2/O_2$ battery (2.81 V) is remarkably higher than that of Super P cathode (2.70 V) at 500 mAh g$^{-1}$. More importantly, the charge process of Li-$CO_2/O_2$ battery can be effectively promoted by the carbon nanosheets catalyst, derived from the much lower charge platform when compared with the Super P based battery. The evident overpotential advantage demonstrates the enhanced catalytic activity promoted by the mesoporous structures, abundant channels, cellular morphology, and sufficient N dopants.

The cycling performance of Li-$CO_2/O_2$ battery with carbon nanosheets and Super P cathode are further investigated for its long-term stability, as shown in Figure 7e. Obviously, the battery with carbon nanosheets cathode delivers noticeable cycling performance of 95 cycles with the lowest voltage gap of 1.21 V. In addition, the discharge and charge platforms are highly stable during the long-term cycling process, which start from the first cycle ($E_{dis}$: 2.82 V, $E_{char}$: 4.03 V) and then slowly increase to the 95 cycles ($E_{dis}$: 2.48 V, $E_{char}$: 4.5 V). In contrast, the battery with Super P cathode under same restricted capacity (500 mAh g$^{-1}$) just sustains 24 cycles with remarkably increased overpotential of 1.75 V. Even under the higher restricted capacity of 1000 mAh g$^{-1}$, the battery with carbon nanosheets cathode can maintain continuous discharge–charge cycling up to 73 cycles with almost flat voltage platform. However, the battery with Super P cathode just cycles 17 times with accelerated battery attenuation. The overpotential and long-term stability

advantages can be attributed to the ordered mesoporous structure, high surface area, and large numbers of defective sites in carbon nanosheets.

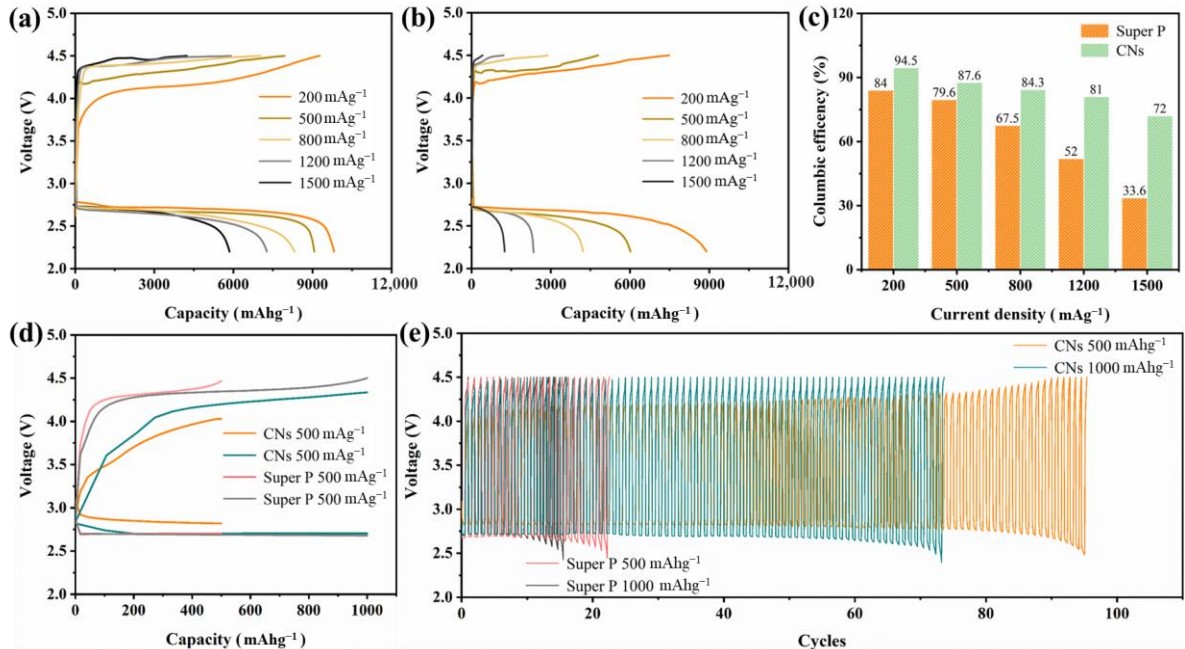

**Figure 7.** Ful discharge–charge curves of Li-$CO_2$/$O_2$ battery with carbon nanosheets (**a**) and Super P (**b**) cathode at different current densities; the corresponding columbic efficiency has been provided for visible comparison (**c**); the first discharge-charge curve (**d**) and cycling performance (**e**) of Li-$CO_2$/$O_2$ battery with carbon nanosheets and Super P cathodes at different restricted capacities and 500 mA g$^{-1}$.

In the Li-$CO_2$/$O_2$ battery, the carbon nanosheets successfully make the neighboring C atoms positively charged and strengthen the $O_2$ adsorption. Especially, the oxygen molecules can directly bond to the lone pair electrons provided by pyridinic-N atoms and tremendously promote the $O_2$ reduction reaction, which is subsequently combined with $CO_2$ and the final discharge products $Li_2CO_3$ forms. Meanwhile, the faster $CO_2$ adsorption kinetics have been accelerated by the N-dopant, which constructs an enriched $CO_2$ atmosphere on the catalyst surface. With the assistance of special cellular morphology and abundant channels effectively improving the $Li_2CO_3$ accommodation, and mesoporous structures largely facilitating the $CO_2$/$O_2$ and ions transfer, the carbon nanosheets successfully fulfill the excellent discharge capacity performance and lower discharge overpotential. During the charge process, the better electronic conductivity and higher surface area can contribute to the faster Li$^+$ and electron diffusion, lower interface current density, and sufficient active sites, which can facilitate the $Li_2CO_3$ decomposition process and thus deliver high cycling stability.

## 3. Materials and Methods

### 3.1. Synthesis of Carbon Nanosheets

The fresh lotus leaves were obtained from a local florist. Before being used, the leaves were washed and ultrasonicated with deionized water several times to remove impurities. The carbon nanosheets were prepared from lotus leaf as follows. Firstly, clean lotus leaf was soaked in ethanol with ultrasonic treatment for 2 h for extracting chlorophyll. After that, the chlorophyll extract was collected and dried in an oven at 80 °C Subsequently, the obtained precursor was pyrolyzed at different temperature (400–800 °C) in $N_2$ atmosphere with different heating rate and calcination time.

### 3.2. Characterization

The X-ray diffraction (XRD) patterns were collected on a X'Pert PRO X-ray Diffractometer with Cu-Kα radiation over a 2θ range of 10–80° (XRD, PANalytical B.V., Almelo, Netherlands). Raman spectroscopy was performed on Lab RAM HR800 (Horiba JobinYvon Co., Ltd., Paris, France) with 532 nm laser. X-ray photoelectron spectroscopy (XPS) was carried on an X-ray photoelectron spectrometer (XPS, ESCA-LAB 250, Kratos, Japan). The morphology characterizations of samples were conducted on a Field emission scanning electron microscope (FE-SEM, FEI, Sirion 200, Eindhoven, The Netherlands). Nitrogen adsorption-desorption isotherms were measured on a Micromeritics ASAP 2020 adsorption analyzer (Micromeritics Co., Ltd., Atlanta, GA, USA). Additionally, the specific surface areas and porosity of the samples were determined with the Brunauer-Emmett-Teller (BET) equation using the Barrett-Joyner-Halenda (BJH) model.

### 3.3. Cathode Preparation and Battery Assembly

During the battery assembly, all the chemical agents were purchased from Aladdin Co., Ltd. (New York, NY, USA). The Li-$CO_2$/$O_2$ batteries were assembled in an argon-filled glovebox with both water and oxygen concentrations less than 0.1 ppm. All those batteries were assembled into CR2025-type coin cells, which consisted of the air cathode, separator (glass fiber, Whatman Co., Ltd., Metstone, UK), electrolyte (1 mol $L^{-1}$ LiTFSI in DMSO), lithium disk, stainless steel spacer, and coin cell cover. Before assembly, LiTFSI was dried in a vacuum oven, and the electrolyte was dried with activated 4 Å molecular sieves, ensuring water concentration less than 20 ppm (measured by Karl Fischer titration). The carbon nanosheets and Super P cathodes were prepared as follows. The mixture composed of carbon nanosheets (Super P) and PVdF (polyvinylidene fluoride, MW 534,000, 99.9%, Sigma Aldrich Co., Ltd., St Louis, MO, USA) at a mass ratio of 9:1, was uniformly dispersed in Nmethyl-2-pyrrolidone (NMP) to form a slurry, which was coated on carbon paper (15.6 × 0.27 mm, Toray Co., Ltd., Tokyo, Japan) and dried at 80 °C overnight in a vacuum oven. The average loading of the active material was adjusted as about 0.4 mg·$cm^{-2}$. In addition, the ratio of $CO_2$/$O_2$ was confirmed as 2:1, as referred to in our previous works [35].

### 3.4. Electrochemical Measurements

The cyclic voltammetry (CV) and electrochemical impedance spectroscopy (EIS) measurements were performed on Zennium IM6 station (Zahner Co., Ltd., Kronach, Germany). For CV experiments, the voltage window was set as 2.0–4.5 V using rotating disk electrode (RDE) with a sweep rate of 1 mV $s^{-1}$. Li metal discs were used as the reference electrode and the counter electrode, glass carbon disc with the diameter of 3 mm as the working electrode, and 1 M LiTFSI in $CO_2$/$O_2$-saturated DMSO as electrolyte. The working electrode consisted of 0.5 mg carbon nanosheets or Super P dispersing in 700 μL of ethanol/DI water (ν:ν = 1:2) solution followed by 10 μL of Nafion solution (5 wt%), which was sonicated for 30 min to prepare a homogeneous ink and then coated on the glass carbon and dried at 80 °C overnight before used. For the EIS measurement, an amplitude of 5 mV within the frequency range of 100 kHz to 10 mHz was applied.

The galvanostatic discharge/charge tests were conducted on a Hantest cycler (Wuhan Hantest Technology Co., Ltd., Wuhan, China) with a voltage range of 2.2–4.5 V (vs. Li/$Li^+$). Before the test, all those batteries were rested for 8 h to reach equilibrium of the oxygen concentrations in the electrolyte. The capacities and current densities of the batteries were calculated based on the mass of the active material (catalyst/Super P).

### 4. Conclusions

In summary, N-doped carbon nanosheets with numerous mesopores and abundant active sites have been successfully synthesized with lout leaf precursor. The combination of ordered channel and pyridinic/pyrrolic nitrogen can improve $CO_2$ adsorption, and the uniform dispersion of abundant graphitic and pyridinic nitrogen is beneficial to the

$O_2$ reduction process, and the pyrrolic nitrogen and enhanced electron conductivity can strengthen the adsorption and transfer of lithium ions and electron, and the porous morphology and high surface area can enlarge the discharge products deposition. Comprehensively, the electrochemical performance of Li-$CO_2$/$O_2$ battery with carbon nanosheets cathode can be dramatically improved based on the underlying mechanism ($4Li^+ + 2CO_2 + O_2 + 4e \rightarrow 2Li_2CO_3$). Accordingly, the battery delivers a high discharge capacity of 9825 mAh $g^{-1}$, acceptable overpotential of 1.21 V, and long-term cycling stability of 95 cycles. The work strongly proves that the biomass-derived carbon materials can be the alternative candidate for high-performance cathode catalyst in Li-$CO_2$/$O_2$ battery. More importantly, multiple advantages of this approach, including, but not limited to, high efficiency and facile preparation, resource-unlimited material, and spontaneous unique morphology, contribute to the development of economic and environmental-friendly Li-$CO_2$/$O_2$ battery.

**Author Contributions:** L.Z.: Supervision, Review and Editing, Data Curation, Funding acquisition. W.K.: Conceptualization, Investigation, Methodology, Writing—Original Draft. L.P.: Investigation, Formal analysis. F.W.: Review and Editing, Funding acquisition, Investigation. All authors have read and agreed to the published version of the manuscript.

**Funding:** This research was funded by National Natural Science Foundation of China (22005227, 51702021), Open Project of Key Laboratory of Green Chemical Engineering Process of Ministry of Education (GCP202118), International Cooperation Foundation of Jilin Province (20220402026GH), and Project of Education Department of Jilin Province (JJKH20210827KJ).

**Data Availability Statement:** The data presented in this study are available on request from the corresponding authors.

**Acknowledgments:** We gratefully appreciate the characterizations from the Analytical and Testing Centre of Huazhong University of Science and Technology, and we appreciate the sample characterization assistance.

**Conflicts of Interest:** The authors declare no conflict of interest.

**Sample Availability:** Samples of the compounds are not available from the authors.

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
