# Peer review of "First Application of Nitrogen-Doped Carbon Nanosheets Derived from Lotus Leaves as the Electrode Catalyst for Li-CO2/O2 Battery"

_catalysts, doi:10.3390/catal13030577_

Round 1
Reviewer 1 Report
In this manuscript, a cellular-like carbon nanosheet catalyst has been synthesized for the Li-CO2/O2 battery. The abundant channel pores of the design catalyst could provide accommodation spaces for the discharge products. The nitrogen dopant, commonly recognized as active sites in carbon, could improve the electron conductivity and accelerate the sluggish ORR/OER kinetics. With these benefits, the battery performance with a low overpotential of 1.21 V and stable cycling performance of 95 cycles has been achieved. Therefore, the reviewer recommends accepting this paper after addressing the following points:
(1) The recent progress about Li-CO2/O2 batteries (DOI: 10.1016/j.ensm.2019.01.009, DOI: 10.31635/ccschem.022.202201876, DOI: 10.1007/s12274-022-4197-6, DOI: 10.1002/anie.202006303) should be added to the introduction part.
(2) The specific surface area is an important information to measure the properties of catalysts. The authors should discuss the surface area variation with carbon nanosheets and super P in detail.
(3) To explain how the oxygen/Li+ circulation was maintained in the coin cells, the authors should supply the photograph of the battery structure.
(4) The DMSO solvent is not stable against Li metal, and thus the by-products will seriously dissipate the oxygen electrochemistry. Therefore, the authors need to explain the special reason for choosing DMSO as the electrolyte solvent.
(5) In Figure 6a, the electrocatalytic activities of the designed catalysts should be discussed according to the CV results.
Reviewer 2 Report
The authors have successfully synthesized the nitrogen-doped carbon nanosheets with enhance electrochemical performance, derived from the louts leaves. The morphology, structure and chemical binding of cellular-like carbon have been discussed and thus catalytic mechanism have been proposed. Moreover, the electrochemical performances of Li-CO2/O2 battery with carbon cathodes have shown the boosting role of the nitrogen doping and cellular morphology in the carbon materials. Overall, this is a good quality paper, a careful study which adds a good knowledge towards enhancing challenging Li-CO2/O2 battery system. Therefore, I think this paper should be published after minor changes, considering my comments and notes.
(1) There is lacking of highlights (3-4 sentences for showing your strength in this manuscript)
(2) The recent advances of Li-CO2(O2) battery research are missing and shall be introduced.
(3) According to the CV in Fig. 6, the authors deduced that all generated superoxide radical are fully consumed by CO2. The related mechanism should be discussed.
(4) The author references mA/g and mAh/g performance values. What is the mass being utilized in this normalization? Mass of carbon nanosheets in electrode, lithium mass? Please state unambiguously
(5) Could you explain how the oxygen circulation was maintained in the coin cells with Li anode if they are sealed? Maybe you could provide a clearer explanation on this in the paper.

Reviewer 3 Report
This manuscript reports that the nitrogen-doped carbon nanosheets has been successfully synthesized with louts leaves and showed enhanced cycle performance in Li-CO2/O2 batteries. The morphology, structure and chemical binding of the cellular-like carbon have been confirmed with SEM, TEM, XRD and XPS. Moreover, the electrochemical performances and impedance spectroscopy of carbon cathodes have shown the boosting role of the nitrogen doping and cellular morphology in the carbon materials. Cycle performances have shown the high retention of the cathode after the charge/discharge procedures. Hence, this manuscript is worth to be published. However more discussions are necessary to make the manuscript clear.
(1) There is lacking of graphical abstract.
(2) The photograph of the coin cells assembly procedure in the electrochemical measurement should be presented.
(3) Is there any full battery test?
(4) In Figure 7, the authors mentioned that the coulombic efficiency decreased with increasing current density. Values of coulombic efficiency obtained at different current densities should be provided.
(5) The authors should provide the equivalent circuit employed, if applicable.
Reviewer 4 Report
The manuscript describes the catalytic nitrogen-doped carbon nanosheets derived from Lotus leaves. Various analyses were conducted from the synthesis of materials to the use of Li-CO2/O2 battery. However, more description and analytical data must be supplemented in the manuscript to prove the author's arguments. Therefore, I recommend this manuscript be published in MDPI after a major revision.
Comment 1:
In Figure 3, the author noted that carbon nanosheets have a higher ID/IG ratio than that of Super P, and the conductivity can be improved. However, in general, graphitization of carbon causes an increase in conductivity. Why did the author argue that the conductivity of carbon nanosheets can be improved by its defect sites?
Comment 2:
The author deconvolute the XPS peak data in Figure 5 d and mentioned three different types of nitrogen. Is there any data that analyzed the formation ratio of each nitrogen and the energy required? Also, it is stated that the benefits vary depending on the type of nitrogen, so please add a reference for this.
Comment 3:
Carbon nanosheets are doped with N without nitrogen precursor or NH3 treatment. Is the lotus leaf already rich in nitrogen?
Comment 4:
The authors call carbon synthesized using lotus leaves N-doped carbon, but XPS data is the only supporting. It would be better to add EDS analysis or elemental analyzer analysis data to support the author’s opinion.
Comment 5:
Is there a reason to set Super P as a control? There will be other biomass-based materials or widely used materials for Li-CO2/O2. There is no reasonable explanation for setting Super P as a control group. Please insert data from other controls and table the performance of the author's material against some references including the references written in the introduction.
Round 2
Reviewer 4 Report
Thank you for your comment and supplementation. The manuscript was faithfully amended in accordance with the comments. It would be better to publish this manuscript in this form.